# Bivalent binding of p14ARF to MDM2 RING and acidic domains inhibits E3 ligase function

Dominika Kowalczyk[1], Mark A Nakasone[1], Brian O Smith[2], Danny T Huang[1,3]

ARF tumor suppressor protein is a key regulator of the MDM2-p53 signaling axis. ARF interferes with MDM2-mediated ubiquitination and degradation of p53 by sequestering MDM2 in the nucleolus and preventing MDM2-p53 interaction and nuclear export of p53. Moreover, ARF also directly inhibits MDM2 ubiquitin ligase (E3) activity, but the mechanism remains elusive. Here, we apply nuclear magnetic resonance and biochemical analyses to uncover the mechanism of ARF-mediated inhibition of MDM2 E3 activity. We show that MDM2 acidic and zinc finger domains (AD-ZnF) form a weak intramolecular interaction with the RING domain, where the binding site overlaps with the E2~ubiquitin binding surface and thereby partially reduces MDM2 E3 activity. Binding of human N-terminal 32 residues of p14ARF to the acidic domain of MDM2 strengthens the AD-ZnF-RING domain interaction. Furthermore, the N-terminal RxFxV motifs of p14ARF participate directly in the MDM2 RING domain interaction. This bivalent binding mode of p14ARF to MDM2 acidic and RING domains restricts E2~ubiquitin recruitment and massively hinders MDM2 E3 activity. These findings elucidate the mechanism by which ARF inhibits MDM2 E3 activity.

## Introduction

The activity of p53 tumor suppressor protein is tightly regulated by MDM2 ubiquitin ligase (E3). Under unstressed, homeostatic conditions, levels of p53 are low because of the E3 activity of MDM2, where MDM2 catalyzes p53 ubiquitination leading to its proteasomal degradation (Kubbutat et al, 1997; Pant & Lozano, 2014). In response to cellular stresses, numerous factors function to uncouple p53 from MDM2 to enable p53 stabilization and activation of its transcriptional activity to restore cellular homeostasis (Hu et al, 2012). In a variety of wild-type p53 cancers, including soft tissues sarcomas, osteosarcomas, and glioblastomas, amplification of the *MDM2* gene has been frequently reported; this drastically reduces p53 protein levels, which in turn promotes cancer growth (Momand

et al, 1998; Stefanou et al, 1998; Burton et al, 2002; Karni-Schmidt et al, 2016). In contrast, deficiency in MDM2 is lethal because of uncontrolled p53 activation as demonstrated by embryonic lethality in mice lacking the *Mdm2* gene (Jones et al, 1995; Montes de Oca Luna et al, 1995). This fine balance between p53 activity and cell fate signifies the importance of precise regulation of MDM2.

MDM2 inhibits p53's transcriptional activity by directly binding p53 and promoting p53 ubiquitination and subsequent degradation by the proteasome (Momand et al, 1992; Oliner et al, 1993; Haupt et al, 1997; Honda et al, 1997; Kubbutat et al, 1997; Shi & Gu, 2012). The importance of MDM2 E3 activity in p53 regulation is underscored by mouse studies, where expression of catalytically inactive *Mdm2* mutants results in embryonic lethality that is rescued by *Trp53* deletion (Itahana et al, 2007; Humpton et al, 2021). The core of MDM2 E3 activity resides in its C-terminal RING domain. The RING domain functions by recruiting E2 thioesterified ubiquitin (E2~Ub; ~indicates thioester bond) and priming the E2~Ub thioester bond for nucleophilic attack by the lysine side chain of substrate (Dou et al, 2012; Plechanovova et al, 2012). MDM2's RING domain activity is regulated by dimerization, where MDM2 RING domain homodimerization or heterodimerization with the MDMX RING domain is essential for binding E2~Ub in the active conformation to confer E3 activity (Nomura et al, 2017; Magnussen et al, 2020). Furthermore, phosphorylation of Ser429 after DNA damage directly contributes to stabilization of E2~Ub to enhance MDM2 E3 activity (Magnussen et al, 2020). In addition to the RING domain, MDM2 consists of three other structured domains connected through regions predicted to be unstructured (Fig 1A). This "beads-on-a-string" architecture enables protein–protein interactions and posttranslational modifications (PTMs) to modulate MDM2 E3 activity (Wade et al, 2013; Fåhraeus & Olivares-Illana, 2014). For example, binding of ribosomal proteins by the zinc finger (ZnF) region inhibits MDM2's activity toward p53 (Lohrum et al, 2003; Zhang et al, 2003; Dai et al, 2004; Dai & Lu, 2004; Jin et al, 2004), and phosphorylation of MDM2 by Chk2, ATM, and CKI promotes MDM2 ubiquitination and degradation (Chen et al, 2005; Pereg et al, 2005; Inuzuka et al, 2010).

The ARF tumor suppressor is one of the most extensively studied negative regulators of MDM2 (Sherr, 2006). Human p14ARF and its close homolog, mouse p19ARF, are reported to interact with MDM2's

[1]Cancer Research UK Beatson Institute, Glasgow, UK   [2]Institute of Molecular Cell and System Biology, University of Glasgow, Glasgow, UK   [3]School of Cancer Sciences, University of Glasgow, Glasgow, UK

Correspondence: d.huang@beatson.gla.ac.uk

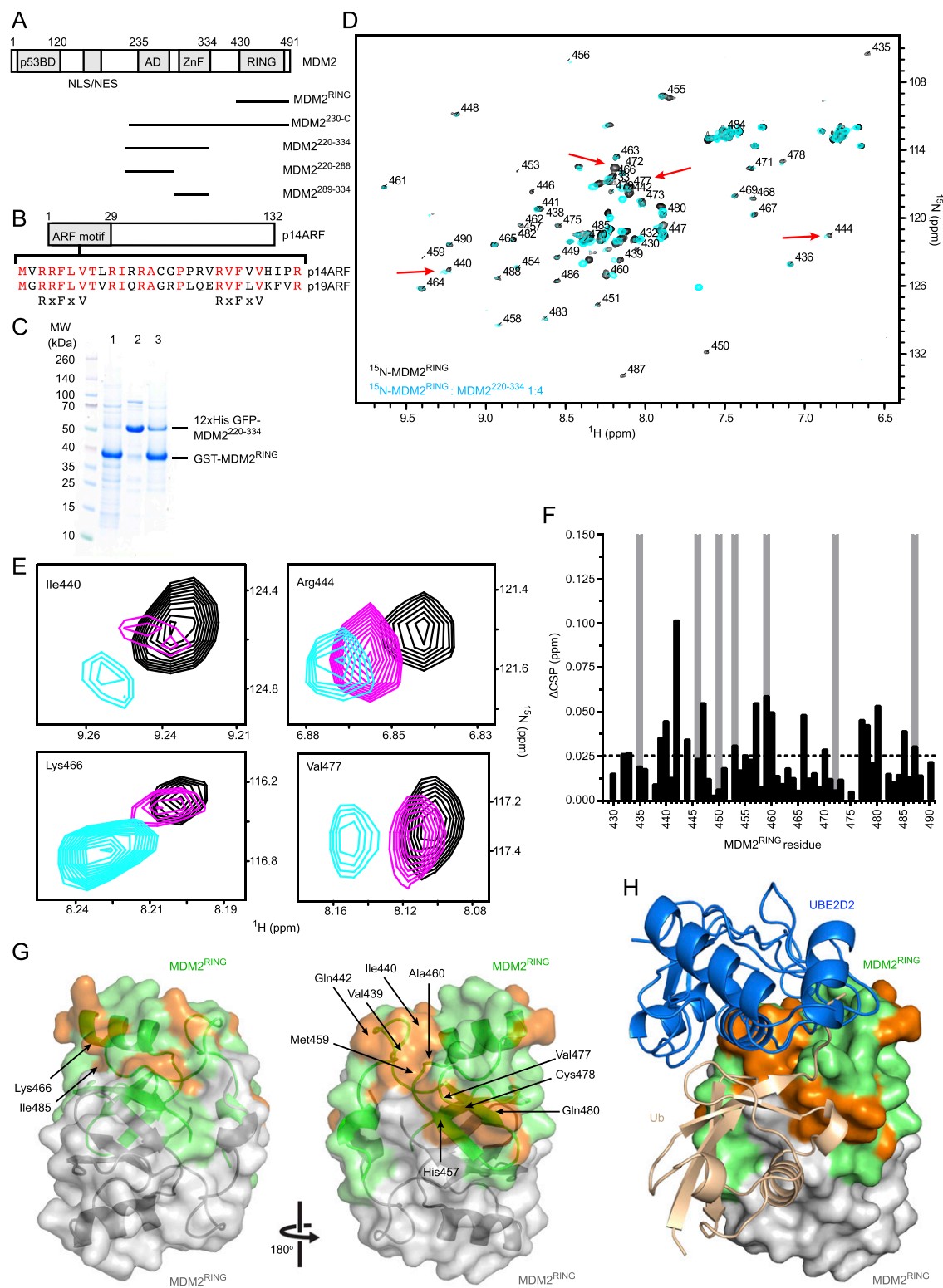

**Figure 1. MDM2$^{220-334}$ interacts with the MDM2$^{RING}$ domain.**
**(A)** Schematic showing the domain structure of MDM2. MDM2 harbors an N-terminal p53-binding domain (p53BD), a region containing nuclear localization sequence and nuclear export sequence (NLS/NES), an acidic and zinc finger (AD-ZnF) region, and a C-terminal RING domain. MDM2 constructs used in this study are indicated. **(B)** Schematic showing the domain structure of human p14ARF. Sequence alignment of the N-terminal 29 residues in p14ARF and p19ARF with conserved residues highlighted in red and ARF motifs (RxFxV) are indicated below. **(C)** SDS–PAGE showing pull-downs of co-expressed GST-MDM2$^{RING}$ and His-GFP-MDM2$^{220-334}$. Lane 1 contains GST-MDM2$^{RING}$ expressed on its own and purified by glutathione affinity chromatography. Co-expression of GST-MDM2$^{RING}$ and His-GFP-MDM2$^{220-334}$ was first

acidic domain (AD), leading to stabilization and activation of p53 (Kamijo et al, 1998; Pomerantz et al, 1998; Stott et al, 1998; Zhang et al, 1998). The N-terminal 60 residues share the highest sequence similarity in p14ARF and p19ARF. Studies showed that the N-terminal 29 residues of p14ARF are sufficient for MDM2 binding and p53 activation (Fig 1B) (Kamijo et al, 1998; Midgley et al, 2000; Lohrum et al, 2000a). This sequence is disordered in solution but forms two extended β-strands upon binding to MDM2 AD (Bothner et al, 2001; DiGiammarino et al, 2001). Importantly, the predicted placement of the β-strands corresponds to the double repetition of the so-called ARF motif, which is a conserved RxFxV sequence present in both homologs of ARF. The mechanism by which the formation of the ARF/MDM2 complex influences p53 has been actively studied over the past two decades. Collectively, studies showed that ARF can localize MDM2 within the nucleolus, thereby separating it from p53, prevent MDM2-mediated p53 nuclear export, and inhibit the E3 activity of MDM2 resulting in p53 stabilization (Honda & Yasuda, 1999; Tao & Levine, 1999; Weber et al, 1999; Zhang & Xiong, 1999; Midgley et al, 2000; Lohrum et al, 2000b; Llanos et al, 2001). In vitro characterization of ARF proteins remains challenging as the N-terminal regions encompassing the first 29 residues of ARF proteins are extremely hydrophobic and basic, whereas also reported to undergo oligomerization and precipitation in aqueous buffers (Bothner et al, 2001; Sivakolundu et al, 2008). Furthermore, protein aggregation prediction (http://old.protein.bio.unipd.it/pasta2/) suggests that the N-terminal 29 residues of ARF proteins have a propensity to form aggregates (Walsh et al, 2014). Thus, it remains difficult to characterize how ARF binding to MDM2 AD could influence MDM2 E3 activity.

A recent study showed that MDM2 could form intramolecular interactions between its AD and RING domains and that this interaction appeared to be enhanced by p14ARF in a pull-down experiment (Cheng et al, 2014). To better understand how p14ARF binding to MDM2 AD could modulate the activity of the RING domain, we focused on the 32 N-terminal residues of p14ARF (N32p14ARF) that is sufficient for MDM2 binding and performed nuclear magnetic resonance (NMR) and biochemical analyses to investigate the interactions between MDM2 AD, MDM2 RING domain, and N32p14ARF. We developed a stable fusion construct containing the N32p14ARF and MDM2 AD-ZnF domains to overcome the challenging behavior of isolated ARF protein. In this study, we show that MDM2 AD-ZnF forms weak interactions with the E2~Ub binding surface of the RING domain to reduce E2~Ub binding affinity and activity. N32p14ARF binding to MDM2 AD-ZnF greatly enhances the interaction with the RING domain and outcompetes E2~Ub to inhibit E3 activity. Moreover, we demonstrate that the N-terminal RxFxV motifs of N32p14ARF are essential for inhibition of MDM2 RING activity. These findings provide insights

into how the ARF tumor suppressor protein inhibits the E3 activity of MDM2 and have implications for therapies directed at MDM2 and how the regulation of MDM2 and p53 contribute to oncogenesis.

# Results

## MDM2 AD-ZnF contacts the E2~Ub binding surface of MDM2 RING domain

A prior study showed that direct fusion of AD (residues 230–260) to the RING domain (residues 410–491) stimulated the catalytic activity of the RING domain (Cheng et al, 2014). Because this fusion construct lacks the region encompassing residues 261–409, how this intramolecular interaction modulates MDM2 activity in the context of native MDM2 sequence remains unclear. To assess interactions between the AD and RING domains, we generated constructs encoding His-GFP–tagged MDM2$^{220-334}$ containing AD and ZnF region (GFP-AD-ZnF) and GST-tagged MDM2$^{419-C}$ containing the RING domain (MDM2$^{RING}$), co-expressed both constructs in *Escherichia coli*, and performed Ni-NTA affinity pull-down followed by glutathione-sepharose affinity pull-down. The double pull-down experiment revealed the formation of a complex between MDM2 AD-ZnF and RING domains (Fig 1C). To confirm this interaction and to map the AD–ZnF–binding site on the RING domain, we purified $^{15}$N-labeled MDM2$^{RING}$ and acquired $^1$H,$^{15}$N-HSQC spectra while titrating MDM2$^{220-334}$ (AD-ZnF). Signals in the $^{15}$N-MDM2$^{RING}$ spectra were assigned based on a previous study (Kostic et al, 2006) (Fig S1A). Titration of $^{15}$N-MDM2$^{RING}$ with a molar excess of AD-ZnF produced moderate chemical-shift perturbations (CSPs) with several attenuations, suggesting a weak interaction between AD-ZnF and the RING domains (Fig 1D–F). The NMR data allowed us to map the AD-ZnF binding surface on the RING domain (Fig 1G). CSPs for many residues near the RING/E2~Ub interface were above average, including Val$^{439}$, Ile$^{440}$, Gln$^{442}$, Met$^{459}$, and Ala$^{460}$, which are involved in binding the hydrophobic region of UBE2D2, and His$^{457}$, Val$^{477}$, Cys$^{478}$, and Gln$^{480}$, which are involved in stabilizing the donor Ub (Nomura et al, 2017; Magnussen et al, 2020) (Fig 1G and H). In addition, several signals from the Lys$^{466}$/Ile$^{485}$ region distal from the E2~Ub binding site were perturbed. To verify that E2~Ub induces CSPs in $^{15}$N-MDM2$^{RING}$, we titrated $^{15}$N-MDM2$^{RING}$ with substoichiometric ratios of UBE2D2–Ub and showed that addition of UBE2D2–Ub led to attenuation of many cross-peaks including those of Ile$^{440}$ and His$^{457}$ along the E2~Ub binding surface, but not Lys$^{466}$ or Ile$^{485}$ (Fig S1B), consistent with the published structure of the MDM2 RING domain bound to UBE2D2–Ub (Magnussen et al, 2020). Together, this demonstrates that AD-ZnF binds to the RING domain and shares a binding surface with E2~Ub.

---

purified by Ni$^{2+}$ affinity chromatography (eluant shown in lane 2) and subsequently purified by glutathione affinity chromatography (eluant shown in lane 3). **(D)** $^1$H,$^{15}$N HSQC spectra of $^{15}$N-MDM2$^{RING}$ alone (black) and in the presence of MDM2$^{220-334}$ at a 1:4 M ratio (cyan). Red arrows indicate the selected close-up view in (E). **(E)** Close-up view of selected $^{15}$N-MDM2$^{RING}$ residues (black) in the presence of MDM2$^{220-334}$ in C, at 1:1 (magenta) and 1:4 (cyan) molar ratios. **(F)** Residue-specific CSPs (black) and signal attenuations (gray) of $^{15}$N-MDM2$^{RING}$ after addition of MDM2$^{220-334}$ at a 1:4 M ratio. Average CSP indicated as a black dashed line. **(G)** Residues with above-average CSP or signal attenuation (orange) from (F) are mapped on the green MDM2$^{RING}$ protomer of MDM2$^{RING}$ homodimer (PDB ID: 6SQO) with the other MDM2$^{RING}$ protomer in gray. Cartoon representation with transparent surface view of MDM2$^{RING}$ homodimer structure is shown. **(H)** The UBE2D2–Ub conjugate (UBE2D2 in blue and Ub in wheat, shown as a cartoon view) is shown in complex with MDM2$^{RING}$ homodimer (PDB ID: 6SQO, shown as a surface view). Colored and oriented as in the right panel of (G).

## AD-ZnF inhibits E3 ligase activity

Given that AD-ZnF and E2~Ub contact the same surface on the RING domain, we set out to determine how AD-ZnF impacts the E3 activity of MDM2. For this purpose, we compared the E3 activity and E2~Ub binding affinity of MDM2[RING] and MDM2[230-C] (containing AD-ZnF and the RING domain). We purified GST-MDM2[RING] and GST-MDM2[230-C] for these analyses (Fig 2A). We performed single-turnover lysine discharge assay by monitoring the rate of UBE2D2~Ub discharge to free lysine in the presence of MDM2 variants. By using saturated lysine as the acceptor, UBE2D2~Ub preferentially transfers Ub to free lysine and eliminates contributions from autoubiquitination arising from differences in lysine residue accessibility in the two MDM2 constructs. We found that MDM2[230-C] displayed reduced activity compared with MDM2[RING] (Fig 2B and C). We also performed surface plasmon resonance (SPR) analysis to measure the UBE2D2–Ub binding affinity. MDM2[RING] exhibited higher affinity for UBE2D2–Ub ($K_d$ = 14.5 ± 1.4 $\mu M$) compared with that of MDM2[230-C] ($K_d$ = 57.5 ± 1.6 $\mu M$) (Fig 2D). These results support the hypothesis that AD-ZnF restricts E2~Ub binding and thereby reduces the E3 activity of MDM2.

To investigate the roles of AD and ZnF in MDM2[RING] binding, we generated GFP-MDM2[220-288] (GFP-AD) and GFP-MDM2[289-334] (GFP-ZnF), where GFP was used to improve the stability of AD and ZnF. We titrated [15]N-MDM2[RING] with molar excess of GFP-AD-ZnF, GFP-AD, or GFP-ZnF. At 1:2 molar ratio of [15]N-MDM2[RING]:GFP-AD-ZnF, several residues exhibited moderate CSPs with several attenuations (Fig S2A), similar to the titration of [15]N-MDM2[RING] with molar excess of AD-ZnF lacking GFP (Fig 1D). In contrast, at the same molar ratio, GFP-AD produced minimal CSPs, whereas GFP-ZnF had no effect (Fig S2B and C). Next, we assessed the effects of GFP-AD-ZnF, GFP-AD, and GFP-ZnF on MDM2[RING]-mediated UBE2D2~Ub discharge. Addition of GFP-AD-ZnF in trans reduced MDM2[RING] activity compared with GFP alone, whereas GFP-AD and GFP-ZnF had no effect (Figs 2E and F and S3). Collectively, these results showed that the combination of AD and ZnF regions are required for binding to the MDM2 RING domain to exert the inhibitory effect.

## The N-terminus of p14ARF interacts with MDM2 AD

The N-terminal region of p14ARF has been shown to bind to the MDM2 AD and reduce MDM2 in vitro autoubiquitination (Midgley et al, 2000). Given that AD-ZnF interacts with the RING domain, we next assessed how p14ARF binding to the AD could modulate the RING domain activity. We verified the interaction between the N-terminal 32 residues of p14ARF (N32p14ARF) and MDM2 by co-expressing His-GFP–tagged N32p14ARF and GST-tagged MDM2[230-C] in *E. coli* and then performing Ni-NTA affinity pull-down followed by glutathione-sepharose affinity pull-down. A complex of His-GFP-N32p14ARF and GST-MDM2[230-C] was detectable after double-affinity pull-down (Fig 3A). We then compared the E3 activity of the His-GFP-N32p14ARF/ GST-MDM2[230-C] complex and GST-MDM2[230-C] using a lysine discharge assay and found that N32p14ARF had an inhibitory effect on the E3 activity (Fig 3B and C). To assess how p14ARF interacts with MDM2, it would be ideal to work with N32p14ARF; however, this construct is not expressed in the soluble form in *E. coli*, and the synthetic peptide does not dissolve in aqueous buffers. Our

double–pull-down result suggests that when N32p14ARF is co-expressed with its binding partner, there is a stabilizing effect. With this in mind, we designed a multidomain fusion protein with GFP-tagged N32p14ARF fused to MDM2[220-334] at the C-terminus (Fig 3D). The resulting GFP-N32p14ARF-[GGSG]₆-MDM2[220-334] fusion protein (hereafter referred to as GFP-N32p14ARF-AD-ZnF) was eluted from the gel filtration column at a volume consistent with a monomer (Fig 3E). Furthermore, the [GGSG]₆ linker was flanked by two thrombin cleavage sites, allowing GFP-N32p14ARF to be freed from AD-ZnF to verify that the two proteins were still bound when no longer fused (Fig 3D–F). With GFP-AD-ZnF and the GFP-N32p14ARF-AD-ZnF fusion, we proceeded to compare how p14ARF-binding to the AD of MDM2 modulates the AD-ZnF inhibition of the RING domain.

## MDM2 AD-ZnF/RING interaction is strengthened by p14ARF

Having shown that AD-ZnF binds MDM2 RING domain and N32p14ARF, we set out to address how E3 activity was affected in the presence of all three components. We titrated the GFP-N32p14ARF-AD-ZnF construct into [15]N-MDM2[RING] and monitored CSPs by [1]H,[15]N-HSQC. At the sub-stoichiometric 1:0.15 molar ratio of [15]N-MDM2[RING]:GFP-N32p14ARF-AD-ZnF, several residues had detectable CSPs, whereas some residues were attenuated (Figs 4A and S4A). There were widespread signal attenuations upon further titration to a stoichiometric molar ratio, indicative of the formation of a larger protein complex leading to enhanced relaxation and line broadening. We mapped the residues with CSPs and attenuated signals obtained at a 1:0.15 molar ratio of [15]N-MDM2[RING]:GFP-N32p14ARF-AD-ZnF onto the structure of the MDM2 RING domain homodimer (Fig 4B). Similar to AD-ZnF, the GFP-N32p14ARF-AD-ZnF fusion affected the E2~Ub binding surface of MDM2 RING domain including above-average CSPs present for Ile[440], His[457], Lys[473], Cys[478], and Gln[480] (Fig 4A–C). In addition, Asn[433] and Glu[436] from the 3[10]-helical turn preceding the RING domain involved in donor Ub binding (Magnussen et al, 2020) displayed above-average CSPs. As with AD-ZnF, few CSPs were observed in the Phe[462]/Ile[485] patch distal from the E2~Ub–binding site (Fig 4B). Overlay of [15]N-MDM2[RING] spectra in the presence of GFP-N32p14ARF-AD-ZnF or GFP-AD-ZnF demonstrates incorporation of N32p14ARF greatly enhances attenuation of the signals indicative of enhanced RING/AD–ZnF interaction (Fig 4D). As the NMR titration experiments were performed with GFP fusions to stabilize N32p14ARF, we verified that GFP alone did not affect signals of [15]N-MDM2[RING] in [1]H,[15]N-HSQC spectra (Fig S4B). To assess whether MDM2[RING] was still active in this ternary complex and with N32p14ARF, we carried out lysine discharge assays. In this assay, discharge of UBE2D2~Ub by MDM2[RING] was monitored across increasing concentrations of GFP-AD-ZnF or GFP-N32p14ARF-AD-ZnF. Addition of GFP-AD-ZnF reduced MDM2[RING] activity compared with GFP alone. In contrast, addition of GFP-N32p14ARF-AD-ZnF further hindered MDM2[RING] activity compared with GFP-AD-ZnF (Figs 4E and S5).

## p14ARF directly binds MDM2 RING to inhibit E3 activity

Thus far, we have shown that N32p14ARF strengthens the MDM2 AD-ZnF-driven inhibition of the RING domain. However, whether this

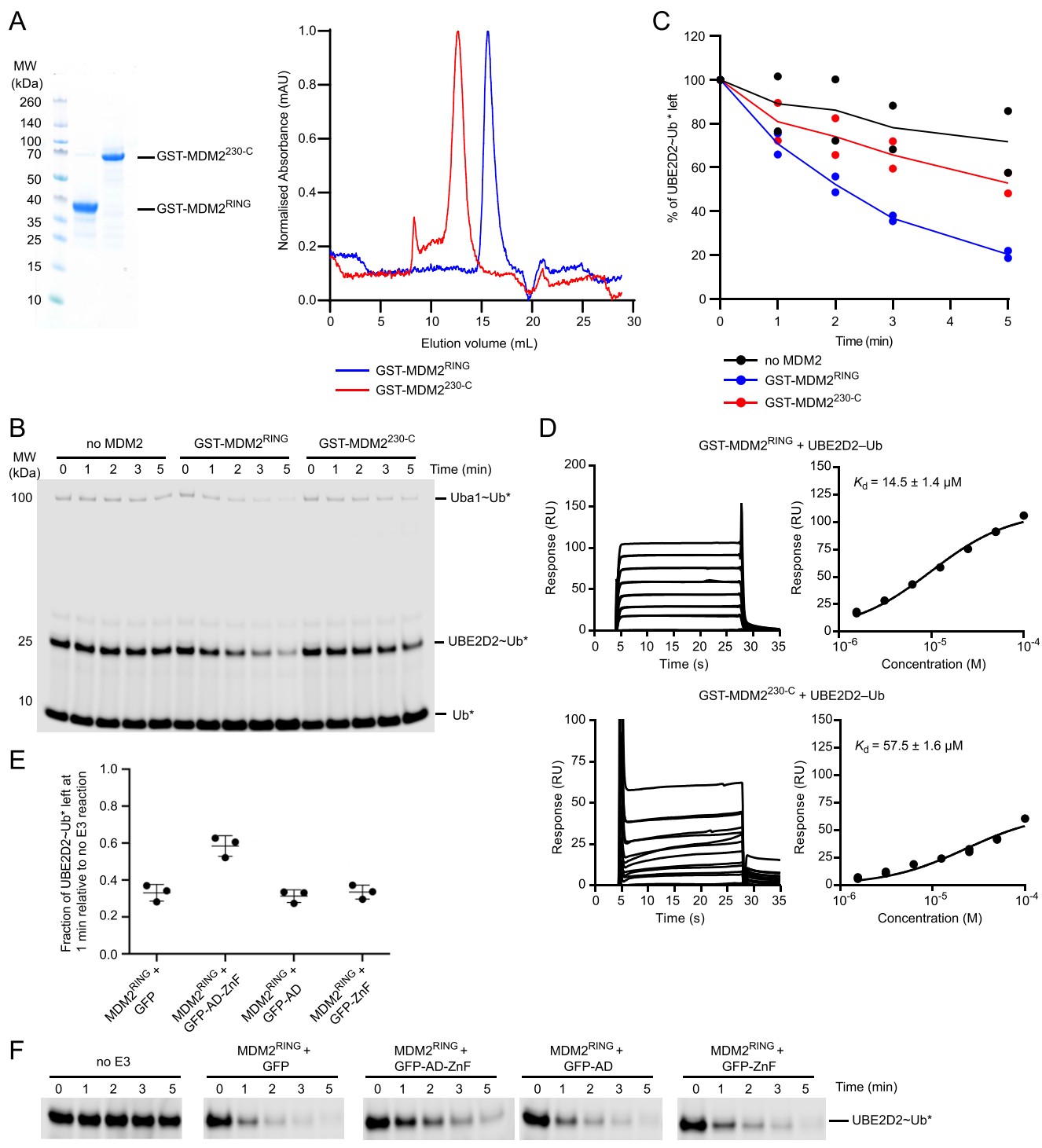

**Figure 2. AD-ZnF of MDM2 inhibits its ligase activity.**
**(A)** SDS–PAGE (left panel) and size exclusion elution profile (right panel) showing the purity of GST-MDM2$^{230-C}$ and GST-MDM2$^{RING}$ proteins. **(B)** Non-reduced SDS–PAGE showing the discharge of UBE2D2~Ub to L-lysine over time, catalyzed by GST-MDM2$^{RING}$ and GST-MDM2$^{230-C}$. Asterisks indicate fluorescently labeled Ub. **(C)** Plot showing the rate of UBE2D2~Ub discharge in (B). Data are from two independent experiments (n = 2). The line represents the mean value. **(D)** Surface plasmon resonance analysis of UBE2D2–Ub binding to GST-MDM2$^{230-C}$ and GST-MDM2$^{RING}$. Sensorgrams are on the left and binding curves on the right. Data are from two independent experiments (n = 2). The equilibrium dissociation constants ($K_d$) are indicated. Error bar indicates SEM. **(E)** Plot of UBE2D2~Ub left at the 1-min time point, corresponding to (F). Data are presented as mean value ± SD from three independent experiments (n = 3). **(F)** Non-reduced SDS–PAGE showing the discharge of UBE2D2~Ub to L-lysine over time, catalyzed by MDM2$^{RING}$ in the presence of 50 μM of GFP, GFP-AD-ZnF, GFP-AD, or GFP-ZnF. Asterisks indicate fluorescently labeled Ub.

**Figure 3. N32p14ARF interacts with MDM2$^{220-334}$.**
**(A)** SDS–PAGE showing pull-downs of co-expressed GST-MDM2$^{230-C}$ and His-GFP-N32p14ARF. Lane 1 contains GST-MDM2$^{230-C}$ expressed on its own and purified by glutathione affinity chromatography. Co-expression of GST-MDM2$^{230-C}$ and His-GFP-N32p14ARF was first purified by Ni$^{2+}$ affinity chromatography (eluant shown in lane 2) and subsequently purified by glutathione affinity chromatography (eluant shown in lane 3). **(B)** Non-reduced SDS–PAGE showing the discharge of UBE2D2-Ub to L-lysine over time, catalyzed by GST-MDM2$^{230-C}$ and the GST-MDM2$^{230-C}$/GFP-N32p14ARF complex. Asterisks indicate fluorescently labeled Ub. **(C)** Plot showing the rate of UBE2D2-Ub discharge in (B). Data are from three independent experiments (n = 3). The line represents the mean value. **(D)** Schematic showing the design of the GFP-N32p14ARF-AD-ZnF construct, with the glycine and serine linker [GGSG]$_6$ flanked by two thrombin cleavage sequences. **(E)** Overlaid size exclusion elution profiles of GFP-N32p14ARF-AD-ZnF alone (green line) and treated with thrombin (magenta line) in comparison to MDM2 AD-ZnF (black line). **(F)** SDS–PAGE showing the eluted peak fraction corresponding to untreated GFP-N32p14ARF-AD-ZnF (lane 1) and thrombin-treated GFP-N32p14ARF-AD-ZnF (lane 2) in (E). Asterisks indicate trace contaminants.

results from direct contact between p14ARF and the RING domain or whether p14ARF-bound AD indirectly modulates the AD-ZnF/RING interaction remains ambiguous. To investigate whether p14ARF interacts directly with MDM2 RING domain, we co-expressed His-GFP–tagged N32p14ARF and GST-tagged MDM2$^{350-C}$ in *E. coli* and performed Ni-NTA affinity pull-down followed by glutathione-sepharose affinity pull-down. GST-MDM2$^{230-C}$ was pulled down together with His-GFP-N32p14ARF after double-affinity pull-down (Figs 3A and 5A). In contrast, His-GFP-N32p14ARF pulled down GST-MDM2$^{350-C}$ on the first Ni-NTA affinity pull-down, but in the second glutathione-sepharose affinity pull-down, only GST-MDM2$^{350-C}$ was present (Fig 5A). We noticed that the His-GFP-N32p14ARF pull-down product was considerably less abundant after the first Ni-NTA affinity pull-down when co-expressed with GST-MDM2$^{350-C}$, suggesting that MDM2 lacking the AD might not be able to make a stable complex with N32p14ARF. Nonetheless, the results suggested that His-GFP-N32p14ARF exhibits weak affinity for MDM2$^{350-C}$. Although

the synthetic N32p14ARF peptide is soluble in DMSO, addition of N32p14ARF peptide to MDM2$^{RING}$ in the aqueous buffer caused precipitation. Thus, to further validate whether p14ARF directly contacts MDM2$^{RING}$, we generated a fusion construct, N32p14ARF-MDM2$^{350-C}$, in which N32p14ARF was linked to a variant of the MDM2 RING domain that included ~80 residues of the native sequence before the RING domain to allow for flexibility and solubility of N32p14ARF (Fig 5B). Discharge of UBE2D2~Ub was reduced with the N32p14ARF-MDM2$^{350-C}$ fusion, suggesting that N32p14ARF has an inhibitory effect on the MDM2 RING domain (Fig 5C). In addition, binding of UBE2D2–Ub to the N32p14ARF-MDM2$^{350-C}$ fusion was abolished (Fig 5D).

Next, we focused on the role of the "ARF motifs" in N32p14ARF in the inhibition of the MDM2 RING domain. Both p14ARF and mouse p19ARF contain two conserved RxFxV motifs within the first and second N-terminal β-strands (Fig 1B). We mutated six key residues in the RxFxV motifs to alanine (R3A,F5A,V7A,R21A,F23A,V25A; 6Ala; Fig

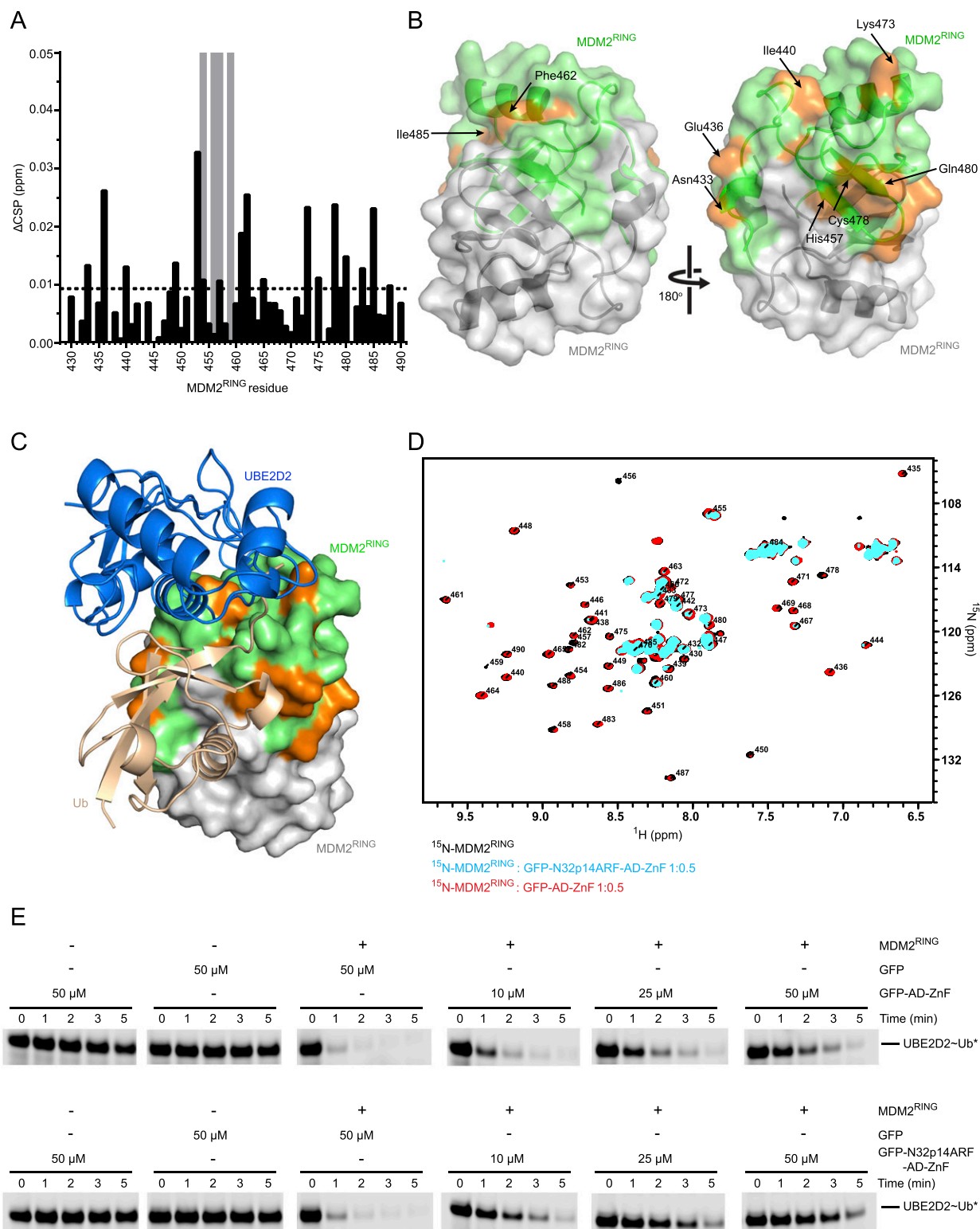

**Figure 4. N-terminus of p14ARF strengthens the MDM2^RING/AD-ZnF inhibitory interaction.**
**(A)** Residue-specific CSPs (black) and signal attenuations (gray) of ^15N-MDM2^RING after addition of GFP-N32p14ARF-AD-ZnF at a 1:0.15 M ratio. The average CSP is indicated by the black dashed line. **(B)** Residues with above-average CSPs or signal attenuation from (A) (orange) are mapped onto the green MDM2^RING protomer in the structure of the MDM2^RING homodimer (PDB ID: 6SQO) with the other MDM2^RING protomer in gray. Cartoon representation with transparent surface view of MDM2^RING homodimer structure is shown. **(C)** UBE2D2–Ub (UBE2D2 in blue and Ub in wheat, shown as a cartoon view) is shown in complex with MDM2^RING homodimer (PDB ID: 6SQO, shown as a surface view). Colored and oriented as in the right panel in (B). **(D)** Overlay of ^1H,^15N HSQC spectra of ^15N-MDM2^RING (black), in the presence of GFP-AD-ZnF (red), and

5B) in the N32p14ARF-MDM2$^{350-C}$ fusion and showed that this N32p14ARF$^{6Ala}$-MDM2$^{350-C}$ fusion had enhanced activity in the UBE2D2~Ub discharge assay and improved UBE2D2–Ub binding affinity compared with the N32p14ARF-MDM2$^{350-C}$ fusion (Fig 5C and D). The N32p14ARF$^{6Ala}$-MDM2$^{350-C}$ fusion was slightly less active and bound UBE2D2–Ub marginally weaker compared with that of MDM2$^{350-C}$, suggesting that other residues in N32p14ARF likely contribute to binding of the RING domain. These data demonstrate that N32p14ARF directly binds and inhibits the E3 activity of the MDM2 RING domain.

# Discussion

Uncovering how MDM2's E3 activity is modulated is critical for understanding how MDM2 regulates p53 levels. The combined NMR and biochemical assays presented in this study demonstrate that the RING domain of MDM2 forms an intramolecular interaction with the AD-ZnF region and recognizes E2~Ub through a conserved surface. Our data support the hypothesis that the AD-ZnF region of MDM2 directly binds the RING domain and restricts the access of E2~Ub, leading to a reduction in E3 activity. NMR and activity assays revealed that the observed AD-ZnF/RING interaction is weaker than the RING/E2~Ub interaction. This suggests that under normal conditions, MDM2 RING domain could still access E2~Ub to exert its E3 activity despite having a slightly reduced E2~Ub binding affinity. Cheng et al (2014) previously reported that the AD could stimulate RING domain activity (Cheng et al, 2014). These differences in the observed effects on RING domain activity may arise from directly fusing the AD to the RING domain in their study, whereas our construct contains the native sequence (MDM2$^{230-C}$) that encompasses the AD-ZnF region connected to the RING domain by a flexible linker. Indeed, our data showed that the combination of AD and ZnF regions are required for optimal AD-ZnF/RING interactions to occlude E2~Ub binding.

Stress signals, such as DNA damage, have been shown to affect MDM2 PTMs, as well as its interactions with binding partners. Therefore, either PTMs or binding partners could potentially modulate the intramolecular interaction between MDM2 AD-ZnF and the RING domain, thereby increasing or abolishing inhibition of MDM2 E3 activity. p14ARF has been shown to bind MDM2 AD, but how it inhibits MDM2 E3 activity remains elusive. We showed that N32p14ARF binding to the AD of MDM2 hindered the activity of the C-terminal RING domain. Furthermore, the N32p14ARF/MDM2 AD-ZnF fusion complex greatly reduced MDM2 E3 activity by blocking the E2~Ub binding surface of the RING domain. The extent of CSPs and signal attenuations in $^{15}$N-MDM2$^{RING}$ spectra and additional CSPs in the 3$^{10}$-helical region in the presence of GFP-N32p14ARF-AD-ZnF compared with GFP-AD-ZnF hinted that N32p14ARF not only strengthens the AD-ZnF and RING domain interaction but also directly binds to the RING domain. Indeed, when N32p14ARF was fused to the N-terminus of MDM2 RING domain, the conserved ARF motifs were essential for inhibiting MDM2 E3 activity. Our data

suggest a model whereby the weak AD–ZnF interaction with the RING domain perturbs E2~Ub binding and allows MDM2 to be "primed" for inhibition (Fig 6). The binding of N32p14ARF to MDM2 AD enhances MDM2 AD–ZnF interaction with the RING domain and enables N32p14ARF to directly engage the RING domain. MDM2 AD-ZnF and the RING domains are separated a long unstructured linker. Therefore, it is likely that N32p14ARF binding would induce a global conformational change in the AD-ZnF and RING domains, restraining the MDM2 RING domain in an autoinhibited conformation and thereby restricting MDM2-mediated ubiquitination.

As documented in over three decades of studies, numerous effectors have been reported to bind to MDM2 (Fåhraeus & Olivares-Illana, 2014). For example, the AD-ZnF region is required for MDM2 interaction with ribosomal proteins, which in turn leads to activation of p53 (Liu et al, 2016). It remains unknown how other effectors of MDM2 exert their inhibitory effect, but it is plausible that they could take advantage of MDM2's "primed" state and inhibit MDM2 through a similar mechanism as ARF. In contrast, when MDM2 binds to p53, MDM2 AD is reported to contact p53 to facilitate MDM2-mediated p53 ubiquitination (Kawai et al, 2003; Meulmeester et al, 2003; Yu et al, 2006). How MDM2 AD facilitates p53 ubiquitination is not clear. It seems likely that MDM2 AD could bind to and orient p53 for optimal ubiquitination by E2~Ub bound to the MDM2 RING domain. At the same time, MDM2 AD binding to p53 would perturb MDM2 AD-ZnF/RING interaction and relieve the inhibition of MDM2 E3 activity to allow MDM2-mediated p53 ubiquitination. p53 has been shown to form a ternary complex with MDM2 and ARF (Kamijo et al, 1998; Stott et al, 1998; Zhang et al, 1998). Although a detailed binding mechanism of the ternary complex is unknown, it seems plausible that ARF would bind the well-established MDM2 AD and thereby uncouples MDM2 AD-p53 contact to perturb MDM2-mediated p53 ubiquitination. Furthermore, based on our findings, we posit that the ARF could engage in bivalent interaction with MDM2 AD and RING domains to restrain MDM2 E3 activity in the ternary complex, resulting in p53 stabilization. Future structural characterization of MDM2/p53/ARF complexes will be required to unveil the detailed binding mechanism.

p14ARF and p19ARF proteins remain challenging to study in vitro because of the amyloid-like feature of their N-terminal sequences (Bothner et al, 2001). The methodology employed to generate stable and monomeric fusions of N32p14ARF and MDM2-AD-ZnF could be useful for future studies in characterizing this complex alone and in the complex with the RING domain at atomic resolution. Structural insight into the binding mode could pave the way for the development of MDM2 RING domain inhibitors that will have paramount clinical importance. Most of the therapeutic strategies focus on the MDM2/p53 interaction by targeting MDM2's N-terminal p53 binding domain with nutlin derivatives or synthetic peptides (Skalniak et al, 2019). Our related work has shown that targeting the RING domain might be a suitable alternative to reactivate p53 (Nomura et al, 2017; Humpton et al, 2021). Therefore, this study will guide future work on MDM2 RING and expands on a promising mechanism to inhibit the E3 activity of MDM2 through the RING domain.

---

GFP-N32p14ARF-AD-ZnF (cyan) with both at a 1:0.5 molar ratio. **(E)** Non-reduced SDS–PAGE showing the discharge of UBE2D2~Ub to L-lysine over time, catalyzed by MDM2$^{RING}$ alone or in the presence of increasing concentrations of GFP-AD-ZnF or GFP-N32p14ARF-AD-ZnF. Asterisks indicate fluorescently labeled Ub.

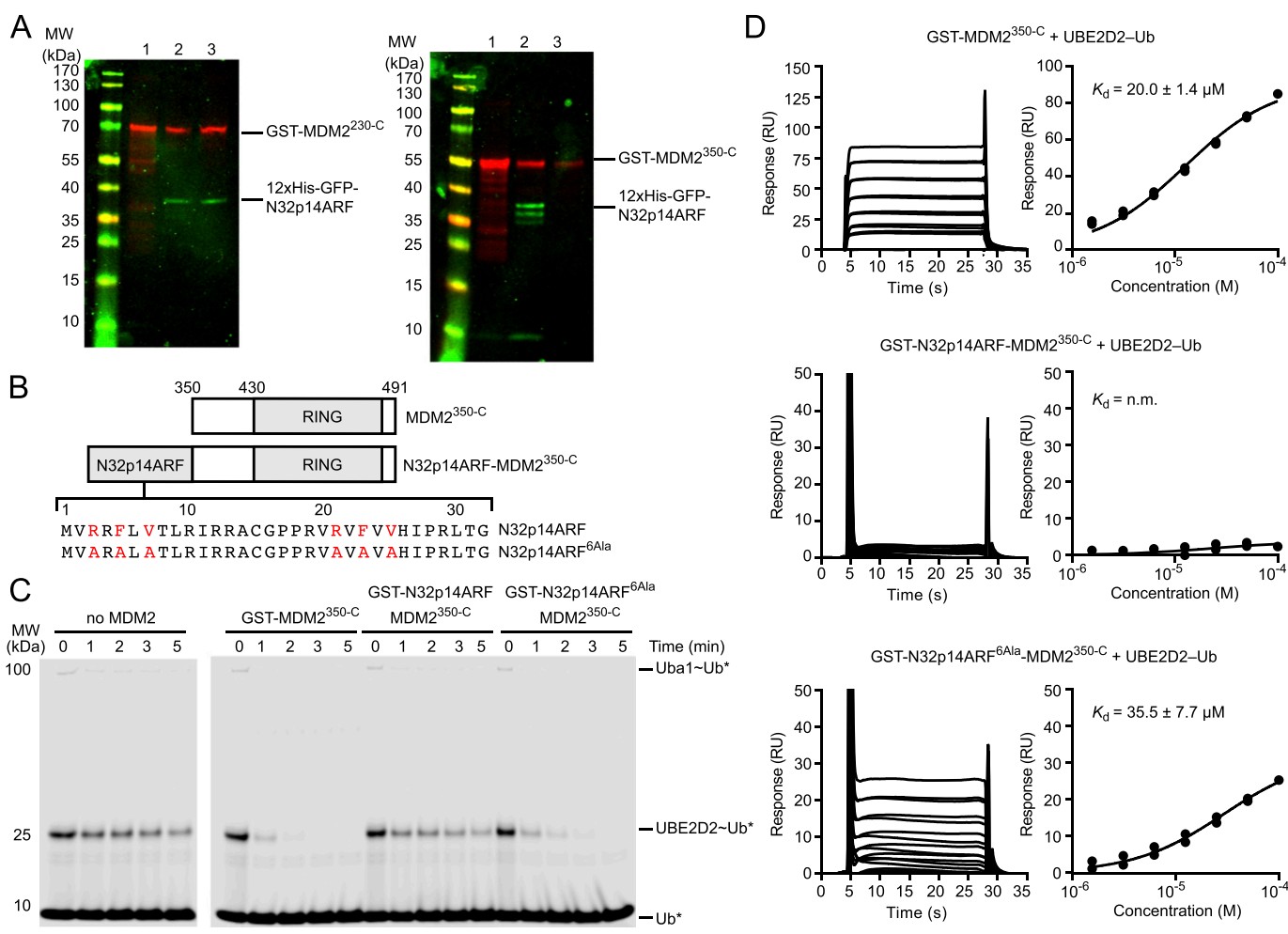

**Figure 5. N-terminus of p14ARF directly targets MDM2$^{RING}$ domain.**
**(A)** Western blot showing pull-downs of co-expressed GST-MDM2$^{230-C}$ (left panel) or GST-MDM2$^{350-C}$ (right panel) and His-GFP-N32p14ARF detected with anti-GST (red) or anti-GFP (green) antibodies. Lane 1 contains GST-MDM2 variants expressed on their own and purified by glutathione affinity chromatography. Lanes 2 and 3 are the eluants from the first (Ni-NTA affinity purification) and second (glutathione affinity purification) steps, respectively. **(B)** Schematic showing the design of the MDM2$^{350-C}$ and N32p14ARF-MDM2$^{350-C}$ constructs. N32p14ARF and N32p14ARF$^{6Ala}$ sequences are shown, where the RxFxV motif is highlighted in red. **(C)** Non-reduced SDS–PAGE showing the discharge of UBE2D2~Ub to L-lysine over time, catalyzed by GST-MDM2$^{350-C}$, GST-N32p14ARF-MDM2$^{350-C}$, and GST-N32p14ARF$^{6Ala}$-MDM2$^{350-C}$. Asterisks indicate fluorescently labeled Ub. **(D)** Surface plasmon resonance analysis of UBE2D2–Ub binding to GST-MDM2$^{350-C}$, GST-N32p14ARF-MDM2$^{350-C}$, and GST-N32p14ARF$^{6Ala}$-MDM2$^{350-C}$. Sensograms are on the left and binding curves on the right. Data are from two independent experiments (n = 2). The equilibrium dissociation constants ($K_d$) are indicated. N.m. indicates not measureable. Error bar indicates SEM.

# Materials and Methods

## Generation of constructs

All DNA constructs were generated using standard PCR techniques with the Q5 High-Fidelity kit (NEB) and verified by automated sequencing. GST-tagged constructs were cloned into a modified pGEX-4T-1 vector (Cytiva), which provides an N-terminal GST sequence followed by a TEV cleavage site. His-tagged constructs were cloned into modified pRSFDuet-1 (Merck Millipore), which provides an N-terminal 6xHis or 12xHis tag, followed by TEV cleavage sequence. MDM2$^{419-C}$ (MDM2$^{RING}$) and MDM2$^{230-C}$ used in the activity assays and MDM2$^{220-334}$ used in the NMR studies were expressed with a GST tag. $^{15}$N-labeled MDM2$^{RING}$ was expressed with a 6xHis tag using a stable variant of MDM2$^{RING}$ harboring G443T (Magnussen &

Huang, 2021). All constructs containing GFP were expressed with a 12xHis tag. The N32p14ARF$^{6Ala}$-MDM2$^{350-C}$ construct was generated by site-directed mutagenesis using PfuUltra High-Fidelity DNA Polymerase (Agilent) and verified by DNA sequencing. The GFP-N32p14ARF-AD-ZnF construct was generated by linking N32p14ARF sequence to GFP via a short GGS linker and further fusing MDM2$^{220-334}$ to the C-terminus of GFP-N32p14ARF via a GLVPRGSGGGGSGGGGSGGGGSG GGSGGGGSGGGSGLVPRGS sequence, where LVPRGS is a thrombin recognition and cleavage site.

## Recombinant proteins expression

All recombinant proteins were expressed in *E. coli* BL21-DE3 (Thermo Fisher Scientific). Cells were grown at 37°C in Luria Bertani medium, until reaching an OD$_{600}$ of 0.6–0.7 and then induced

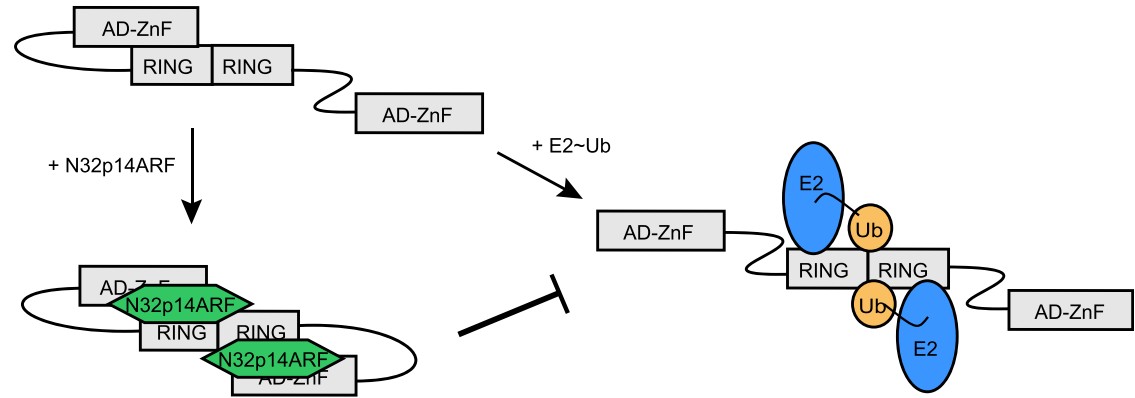

**Figure 6. Model of N32p14ARF-driven inhibition of MDM2 E3 activity.**
MDM2 AD-ZnF exhibits weak binding for the E2~Ub binding surface of the RING domain and partially perturbs MDM2 E3 activity. Binding of N32p14ARF to the AD region of MDM2 likely induces a global conformational change, where N32p14ARF strengthens the MDM2 AD–ZnF interaction with its RING domain and contacts the MDM2 RING domain to restrict MDM2 E3 activity. For simplicity, only the AD-ZnF and RING domains of MDM2 are shown.

with 0.2 mM isopropyl $\beta$-D-1-thiogalactopyranoside (IPTG) at 20°C overnight. $^{15}$N-MDM2$^{RING}$ was obtained from M9 medium based on a previously published protocol (Studier, 2005). In summary, 300 ml of Luria Bertani medium was inoculated from a single colony and grown overnight at 37°C. Cells were pelleted, washed in M9 medium, and resuspended to a final volume of 60 ml. Cells were then transferred to 6 liters of M9 medium containing $^{15}$NH$_4$Cl (1 g/l) and grown at 37°C until reaching an OD$_{600}$ of 0.6–0.7 and induced with 0.2 mM IPTG at 16°C for 20 h.

**Protein purification**

Bacterial cells were centrifuged and lysed with a microfluidizer. Cells expressing His-tagged constructs were resuspended in buffer containing 25 mM Tris–HCl (pH 7.6), 400 mM NaCl, 1 mM Tris(2-carboxyethyl)phosphine (TCEP), 5% (vol/vol) glycerol, and imidazole (25 mM for 6xHis tag protein and 50 mM for 12xHis tag protein). Cells expressing GST-tagged constructs were resuspended in buffer containing 50 mM Tris–HCl (pH 7.6), 400 mM NaCl, 1 mM TCEP, and 5% (vol/vol) glycerol. The same buffers were used for washing during the affinity chromatography steps. All proteins, except for $^{15}$N-MDM2$^{RING}$, were purified into a final buffer containing 25 mM Tris–HCl (pH 7.6), 400 mM NaCl, and 1 mM TCEP. The buffer used for elution of GST-tagged proteins contained 50 mM Tris–HCl (pH 8.0), 400 mM NaCl, 1 mM TCEP, 5% (vol/vol) glycerol, and 10 mM glutathione (GSH), whereas buffer used for elution of His-tagged proteins contained 25 mM Tris–HCl (pH 7.6), 400 mM NaCl, 1 mM TCEP, 5% (vol/vol) glycerol, and 200 mM imidazole. GST-tagged MDM2 variants were purified by glutathione affinity chromatography, followed by HiLoad 26/600 Superose6 chromatography. GST-N32p14ARF-MDM2$^{350-C}$ variants were purified by glutathione affinity chromatography, and buffer exchanged into the final buffer. MDM2$^{220-334}$ was purified by glutathione affinity chromatography, followed by cleavage from the resin with TEV protease at 4°C overnight. Cleaved MDM2 was collected and purified by HiLoad 26/600 Superose6 chromatography. His-GFP-MDM2$^{220-334}$, His-GFP-MDM2$^{220-288}$, His-GFP-MDM2$^{289-334}$, and His-GFP-N32p14ARF-AD-ZnF were purified by Ni-NTA affinity chromatography, followed by

HiLoad 26/600 Superose6 chromatography. MDM2$^{RING}$ was purified by Ni-NTA affinity chromatography, followed by buffer exchange on a HiPrep 26/10 Desalting column. The His tag was removed by incubation with TEV protease at 4°C overnight, followed by Ni-NTA pass back. Cleaved MDM2$^{RING}$ was purified by HiLoad 26/600 Superdex 75 chromatography. $^{15}$N-MDM2$^{RING}$ was purified by Ni-NTA affinity chromatography without removal of the 6xHis-tag, followed by HiLoad 26/600 Superdex 75 chromatography into buffer containing 20 mM sodium phosphate, (pH 7.0), 300 mM NaCl, and 1 mM TCEP. Protein concentration was determined by Bio-Rad protein assay. His-GFP-MDM2$^{220-334}$, His-GFP-MDM2$^{220-288}$, His-GFP-MDM2$^{289-334}$, and His-GFP-N32p14ARF-AD-ZnF were determined by using molar extinction coefficient at 280 nm. MDM2 protein concentration was determined based on the molecular weight of a monomer.

**In vitro pull-down experiments**

GST-MDM2$^{RING}$ and 12xHis-GFP-MDM2$^{220-334}$ (Fig 1C), GST-MDM2$^{230-C}$ and 12xHis-GFP-N32p14ARF (Figs 3A and 5A), or GST-MDM2$^{350-C}$ and 12xHis-GFP-N32p14ARF (Fig 5A) were co-expressed in *E. coli*. Six liters of cells were grown according to the aforementioned recombinant protein expression. Cells were centrifuged and lysed by sonication. Clarified lysates were applied onto 400 $\mu$l of Ni-NTA resin, washed, and eluted with the aforementioned Ni-NTA affinity purification buffers. The eluted fraction was then mixed with 200 $\mu$l of GSH resin at 4°C for 1 h, washed, and eluted with aforementioned GSH-affinity purification buffers. Total protein was quantified by Bio-Rad protein assay, and 5 $\mu$g of the eluted fraction was analyzed by SDS–PAGE and detected with InstantBlue staining (Figs 1C and 3A), anti-GST (Cytiva 27-4577-01; 1:1,000 dilution), and IRDye 800CW donkey anti-goat IgG (LI-COR Biosciences; 1:15,000 dilution) antibodies or anti-GFP (Santa Cruz 81045; 1:1,000 dilution) and IRDye 800CW goat anti-mouse IgG (LI-COR Biosciences; 1:15,000 dilution) antibodies (Fig 5A).

**SPR assays**

All experiments were done at 25°C using a Biacore T200, equilibrated in running buffer containing 25 mM Tris–HCl (pH 7.6), 300 mM

NaCl, 0.5 mM TCEP, and 0.005% (vol/vol) Tween-20. CM-5 chips with anti-GST-nanobodies (GST VHH; Chromotech) were used to couple GST-MDM2 variants to a level of ~500 response units, with GST as a control. UBE2D2$^{S22R\ C85K}$–Ub (UBE2D2–Ub), used as an analyte, was purified as described previously (Buetow et al, 2015) and serially diluted in running buffer. Binding between GST-MDM2 variants and UBE2D2–Ub was measured in duplicate, across seven concentrations of UBE2D2–Ub, starting at 100 $\mu$M and decreasing in a twofold manner. Data shown are the difference between SPR signal recorded for GST-MDM2 variants and GST alone. The data were analyzed by steady-state affinity using Biacore T200 evaluation software (GE Healthcare) and plotted in Prism8 (GraphPad).

### Single-turnover lysine discharge assays

Lysine discharge assays were performed as described previously (Buetow et al, 2015). *Arabidopsis thaliana* UBA1, UBE2D2$^{S22R}$, and IRDye 800CW maleimide-labeled GGSC-Ub were prepared as described previously (Magnussen et al, 2020). Briefly, 2 $\mu$M of UBE2D2$^{S22R}$ was charged with 4 $\mu$M of maleimide-labeled Ub in the presence of 0.2 $\mu$M *A. thaliana* UBA1 in buffer containing 50 mM Tris–HCl (pH 7.6), 50 mM NaCl, 5 mM MgCl$_2$, and 20 mM ATP for 20 min at room temperature and stopped by addition of 25 mM EDTA for 5 min. Lysine (final concentration of 250 mM for Fig 2B; 50 mM for Figs 2F, 3B, 4E, and 5C) and the indicated MDM2 variant or MDM2/p14ARF complex (final concentration: 100 nM for Fig 2B; 200 nM for Figs 2F, 3B, 4E, and 5C) were then added to initiate the reaction. Because of the nanomolar concentration of MDM2 variants used in the assay, it was not possible to visualize MDM2 variant bands by Coomassie staining. To standardize the MDM2 variant concentration used in the assay, we determined the MDM2 variants protein concentration and then loaded equimolar quantities on SDS–PAGE to ensure similar loading of MDM2 variants (Fig S6) before dilution for use in the assay. For Fig 4E, indicated concentrations of 12xHis-GFP-MDM2$^{220-334}$ or 12xHis-GFP-N32p14ARF-AD-ZnF were incubated with MDM2$^{RING}$ for 30 min at room temperature before addition to the charged UBE2D2~Ub to initiate the reaction. Reactions were quenched with SDS loading buffer at the indicated times, analyzed by SDS–PAGE and visualized using a LI-COR Odyssey CLx scanner. Images were prepared using Image Studio software (LI-COR Biosciences). The reactions in Figs 2B and 3B were performed in duplicate and triplicate, respectively. Image Studio software (LI-COR Biosciences) and ImageJ were used to analyze the scanned gels and quantify the intensities of the bands. The plots were generated using Prism8 (GraphPad).

### Size exclusion analysis

Size exclusion analyses in Fig 2A were done using the 10/300 Superdex 200 chromatography column. 500 $\mu$l of protein (~200–500 $\mu$g) was applied onto the column pre-equilibrated in buffer containing 25 mM Tris–HCl (pH 7.6), 400 mM NaCl, and 1 mM TCEP. Data were presented using wavelength at 215 nm and normalized for comparison. To demonstrate that N32p14ARF interacts with MDM2$^{220-334}$ in the GFP-N32p14ARF-AD-ZnF (Fig 3E), 300 $\mu$g of GFP-N32p14ARF-AD-ZnF was incubated in either buffer alone or in

the presence of 3 $\mu$g of thrombin at 4°C overnight before loading onto the 10/300 Superdex 200 column.

### Solution NMR experiments

All NMR experiments were acquired using a Bruker Avance III 600-MHz spectrometer equipped with a cryogenic triple resonance inverse probe. Before titration, each sample was exchanged into NMR buffer (20 mM sodium phosphate, 300 mM NaCl, 0.5 mM TCEP, 5.0% [vol/vol] D$_2$O, pH 7.0). Experiments were carried out at 298 K, and $^1$H-$^{15}$N HSQC spectra were recorded with 16 scans using 64 complex points with a sweep width of 36 parts per million (ppm) in the $^{15}$N dimension. All spectra were processed with 256 points in the indirect dimension using Bruker TopSpin version 3.5 patch level 7 and analyzed using CARA (http://cara.nmr.ch/) and CCPnmr (Skinner et al, 2016).

CSPs were calculated following:

$$CSP = \left[ (\delta_{HA} - \delta_{HB})^2 + ((\delta_{NA} - \delta_{NB})/5)^2 \right]^{1/2}$$

For a given residue in state A or B, $\delta_H$ and $\delta_N$ are the differences in proton and nitrogen chemical shifts, respectively.

## Data Availability

The data supporting the findings of this study are within the paper and its supplemental information.

## Supplementary Information

## Acknowledgements

We would like to thank the Core Services and Advanced Technologies at the Cancer Research UK Beatson Institute (C596/A17196 and A31287), Catherine Winchester, and Lori Buetow for critical reading of the manuscript. This work was supported by Beatson Institute core funding from Cancer Research UK (A17196 and A31287), Cancer Research UK funding to DT Huang (A23278/A29256), and the European Research Council (ERC) under the European Union's Horizon 2020 research and innovation programme (grant agreement no 647849).

### Author Contributions

D Kowalczyk: conceptualization, data curation, formal analysis, investigation, methodology, and writing—original draft, review, and editing.
MA Nakasone: data curation, formal analysis, investigation, methodology, and writing—original draft, review, and editing.
BO Smith: data curation, formal analysis, and writing—review and editing.

DT Huang: conceptualization, formal analysis, supervision, funding acquisition, investigation, project administration, and writing—original draft, review, and editing.

## Conflict of Interest Statement

DT Huang is a consultant for Triana Biomedicines. The other authors declare no competing interests.

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
