## [Reviewer comments · Life Science Alliance]

Life Science Alliance

Bivalent binding of p14ARF to MDM2 RING and acidic domains inhibits E3 ligase function

Danny Huang, Dominika Kowalczyk, Mark Nakasone, and Brian Smith

DOI: <https://doi.org/10.26508/lsa.202201472>

Corresponding author(s): *Danny Huang, Cancer Research UK Beatson Institute*

Review Timeline:

Submission Date:	2022-04-04
Editorial Decision:	2022-05-05
Revision Received:	2022-06-24
Editorial Decision:	2022-07-12
Revision Received:	2022-07-19
Accepted:	2022-07-19

Scientific Editor: Novella Guidi

Transaction Report:

May 5, 2022

Re: Life Science Alliance manuscript #LSA-2022-01472-T

Prof. Danny T Huang
Beatson Institute for Cancer Research
Glasgow, Scotland
United Kingdom

Dear Dr. Huang,

Thank you for submitting your manuscript entitled "Bivalent binding of p14ARF to MDM2 RING and acidic domains inhibits E3 ligase function" to Life Science Alliance. The manuscript was assessed by expert reviewers, whose comments are appended to this letter. We invite you to submit a revised manuscript addressing the Reviewer comments.

Thank you for this interesting contribution to Life Science Alliance. We are looking forward to receiving your revised manuscript.

Sincerely,

B. MANUSCRIPT ORGANIZATION AND FORMATTING:

Reviewer #1 (Comments to the Authors (Required)):

The manuscript by Kowalczyk and colleagues describes a biochemical analysis of the mechanism of MDM2 inhibition by the ARF tumor suppressor. Regulation of MDM2 ubiquitin E3 ligase activity by phosphorylation and protein interactions mediate stress-induced accumulation of p53 tumor suppressor. ARF is a major regulator of the p53 pathway, but its mechanism of action has been challenging to study at the biochemical level due to tendency to form protein aggregates in complex with MDM2.

The authors investigated the effect of ARF on MDM2 acidic domain-RING domain interaction and E3 ligase activity using NMR, SPR, copurification, and ubiquitin release assays. The study expanded on previous report of MDM2 intramolecular interaction, the results provided novel structural and mechanistic insight on ARF regulation of MDM2. Their results suggest that intramolecular interaction between AD-ZF and RING domains of MDM2 results in modest inhibition of RING E3 ligase activity by competing with RING binding to charged E2, and inhibiting the ability of RING to stimulate ub release. Attaching ARF N terminal region to AD-ZF stabilizes AD-RING interaction and inhibits RING E3 activity. Furthermore, ARF-RING direct interaction was detected using copurification assay, tethering ARF to RING also inhibited RING E3 ligase activity. The results led to a model in which ARF binding to AD stabilizes AD-RING interaction, ARF also forms a direct interaction with RING to facilitate the inhibition of RING E3 function.

Comments:

The authors successfully addressed some technical challenges in working with ARF by tethering ARF fragment to MDM2 AD and RING domains. Despite potential caveats with this approach, the results described in the manuscript provide important mechanistic insight on how ARF regulates MDM2 and activates p53. The model is rational and may also be instructive in elucidating how ribosomal proteins activate p53 by binding to MDM2. However, the model is likely a simplified version of the actual process since AD also interacts with p53 core domain, and p53 ubiquitination was not used as readout of E3 activity.

Previous studies showed the MDM2 AD was needed for ubiquitinating p53 in vivo, suggesting it stimulates RING E3 activity or has other positive effects. The authors may want to comment on this discrepancy with their model, given the complexity of the in vivo reactions. Ideally the roles of AD and ZF could be further examined using their ub release assay, for example Figure 2B could include additional truncation mutant that sequentially remove the AD and ZF. Details of how the ARF fragment engage in bivalent binding to AD and RING could also be discussed in more detail, such as whether ARF using separate motifs to bind AD and RING. It would be nice to have reciprocal HSQC NMR mapping of sites on the AD-ZF region that interact with RING domain, the authors may want to comment on whether this was attempted.

Reviewer #2 (Comments to the Authors (Required)):

The manuscript by Kowalczyk et al (from the laboratory of Danny Huang) focuses on understanding the mechanism by which p14ARF regulates the E3 ligase activity of MDM2. In many respects this is a challenging project as some of the proteins are poorly behaved and the interactions are relatively weak. However, the data are generally well presented and will be of interest to others.

In this manuscript the authors use biochemical approaches to investigate how the E3 ligase of MDM2 is modulated, with a focus on the activation domain (AD) and how this modulates RING domain dependent activity. In particular, they examine how the N-terminus of p14ARF influences activity. This paper extends previous data that suggests the reported interactions are likely, but also suggests a molecular mechanism by which p14ARF regulates activity. Notably they present data that suggests that p14ARF stabilises an interaction between the AD and the RING domain of MDM2, which blocks E2~Ub binding and inhibits activity. There are however, some gaps - these are mostly clearly stated although some revision of the Discussion may help make this clearer.

Points to consider:

- 1) The authors should review the section about ARF in the introduction and see if there are ways to improve clarity.
- 2) The use of 're-localize' should be reviewed (note there are no page numbers!)

- 3) The last section of the Introduction should be revised.
- 4) The MDM constructs used in this study should be clearly indicated in Figure 1A.
- 5) Figure 2B legend and related text needs to be revised. There is discussion of co-expression in the text but lane 1 indicates GST-MDM2 is expressed on its own. The origin of lanes 2&3 is not clear. A flow diagram might help.
- 6) It would be good to indicate the regions of the HSQC spectrum in Figure 1C that are blown up in 1D.
- 7) Figures 1F & G need to be improved as the RING dimer is not obvious relative to structural features. It might be good to include a ribbon diagram, with a transparent surface (or something similar) to help orient the reader.
- 8) In figure 2 and the related text it is stated that the AD competes for the E2 binding site. It is clear that the AD restricts binding but not that it competes for the same binding site. Additional data should be provided to support this statement, or the text revised.
- 9) The text/legend related to Figure 3A should be revised in a similar way as outlined for Figure 2B. Currently it is confusing.
- 10) In Figure 3B the difference in activity seems modest, it would be good if this experiment was repeated and the data quantified, much like Figure 2C.
- 11) The results presented in Figure 3D/E suggest quite a tight interaction given the linker was cleaved. This is a nice result!
- 12) The structural data presented in Figure 4 might also warrant revision of the figures as indicated above.
- 13) It may be possible to improve/simplify the labelling of Figure 4E.
- 14) In relation to figure 5c it is stated that when fused to p14ARF E2~ub discharge by MDM2 is nearly 'abolished'. While slower, it appears that by 5 minutes 95% of the conjugate is discharged. The authors should more accurately represent the difference in activity. It might also be useful to number the lanes and refer to them directly.
- 15) It would be helpful to enhance figure 5B so that the different constructs are more obvious - currently the domain diagram at the top is not well integrated.
- 16) The Discussion is quite long and in places fairly speculative. The authors should review with the goal of distinguishing models from established facts.

We thank reviewers for their comments. Below we address the comments in blue.

Reviewer 1

The authors successfully addressed some technical challenges in working with ARF by tethering ARF fragment to MDM2 AD and RING domains. Despite potential caveats with this approach, the results described in the manuscript provide important mechanistic insight on how ARF regulates MDM2 and activates p53. The model is rational and may also be instructive in elucidating how ribosomal proteins activate p53 by binding to MDM2. However, the model is likely a simplified version of the actual process since AD also interacts with p53 core domain, and p53 ubiquitination was not used as readout of E3 activity.

We thank the reviewer for the positive comments. In the discussion we have commented on interplay between MDM2 AD, ARF and p53 (see below).

Previous studies showed the MDM2 AD was needed for ubiquitinating p53 in vivo, suggesting it stimulates RING E3 activity or has other positive effects. The authors may want to comment on this discrepancy with their model, given the complexity of the in vivo reactions.

The AD stimulation of MDM2 E3 activity was based on a prior study (Cheng *et al.* MCB 2014) where AD (residues 230-260) was directly fused to the MDM2 RING domain (residue 410-491). This fusion construct seemingly enhances the discharge of E2~Ub leading to proposed AD stimulation of E3 activity. Its noteworthy that this fusion construct lacks ZnF domain and ~100 linker sequence and therefore may not be representative of how MDM2 AD functions. We showed that AD-ZnF region binds to the RING domain and inhibit the E3 activity. We have discussed this difference in the results and discussion. In the revised manuscript, we further showed that both AD and ZnF are important for binding and inhibition of MDM2 E3 activity.

Prior studies have shown that MDM2 AD is required for ubiquitinating p53 in cells. Most of the experiments were done by deleting MDM2 AD, which resulted in the decrease in p53 ubiquitination. Subsequent in vitro biophysical studies showed that MDM2 AD could bind p53 DNA-binding domain (DBD). Based on these findings, it seems plausible that MDM2 AD might facilitate p53 binding in addition to the N-terminal p53-binding domain. By binding to both p53 N-terminus and DBD, MDM2 might arrange p53 in a configuration that is favorable for ubiquitination by the E2~Ub bound to MDM2 RING domain. ARF binding to MDM2 AD could perturb MDM2 AD-p53 interaction and reduce p53 ubiquitination. We have now revised this in the discussion and noted that this is just a speculation and will require future structural insight into p53-MDM2 and p53-MDM2-ARF complexes to unveil the exact mechanism.

Ideally the roles of AD and ZF could be further examined using their ub release assay, for example Figure 2B could include additional truncation mutant that sequentially remove the AD and ZF. Details of how the ARF fragment engage in bivalent binding to AD and RING could also be discussed in more detail, such as whether ARF using separate motifs to bind AD and RING. It would be nice to have reciprocal HSQC NMR mapping of sites on the AD-ZF region that interact with RING domain, the authors may want to comment on whether this was attempted.

We thank the reviewer for the suggestion. We did not attempt the reciprocal HSQC NMR experiment. Prior studies by Kriwacki's lab showed that MDM2 AD (residues 210-304) is intrinsically unstructured in solution and therefore is not feasible to assign the cross-peaks to gain structural insight. Furthermore, the interaction between AD-ZnF and MDM2 RING domains is weak, and we only observed changes in CSPs in ^{15}N -MDM2^{RING} when AD-ZnF was titrated at molar excess. To perform the reciprocal experiment, one would require to titrate highly concentrated MDM2^{RING}. From our experience MDM2^{RING} has tendency to aggregate and precipitate when concentrated to above 10 mg/ml and therefore we did not consider the reciprocal experiment. We have now generated GFP-AD and GFP-ZnF and assessed their effects on E2~Ub discharge assay and binding to ^{15}N -MDM2^{RING} in comparison to GFP-AD-ZnF. GFP tag was included to improve the stability of the fragments and purification. We showed that titration of GFP-AD-ZnF to ^{15}N -MDM2^{RING} at 2:1 molar ratio caused changes in CSPs. In contrast, titration of GFP-AD had minor effect and GFP-ZnF had no effect. Furthermore GFP-AD-ZnF reduced the MDM2^{RING} activity in E2~Ub discharge assay but not GFP-AD or GFP-ZnF. Our data suggest that both AD and ZnF contribute to MDM2^{RING} binding. This data is now included in Figure 2E/2F and Supplementary Figure 2.

We do not know how ARF fragment engages in bivalent binding to AD and RING domain. Kriwacki's lab has applied NMR to characterize how N-terminal ARF peptide (2-10) binds to MDM2 AD (Sivakolundu *et al*, JMB 2008). Due to the challenges in the solubility and oligomerization of ARF peptide, they were unable to gain any detailed structural insight. Likewise we have experience difficulties with the isolated N32p14ARF peptide. The N32p14ARF-MDM2 AD-ZnF fusion construct provided in our manuscript could serve as a starting point for future structural characterization of ARF-MDM2 complex. We are trying our endeavor to purify sufficient quantity for structural characterization.

Reviewer 2

The manuscript by Kowalczyk et al (from the laboratory of Danny Huang) focuses on understanding the mechanism by which p14ARF regulates the E3 ligase activity of MDM2. In many respects this is a challenging project as some of the proteins are poorly behaved and the interactions are relatively weak. However, the data are generally well presented and will be of interest to others.

In this manuscript the authors use biochemical approaches to investigate how the E3 ligase of MDM2 is modulated, with a focus on the activation domain (AD) and how this modulates RING domain dependent activity. In particular, they examine how the N-terminus of p14ARF influences activity. This paper extends previous data that suggests the reported interactions are likely, but also suggests a molecular mechanism by which p14ARF regulates activity. Notably they present data that suggests that p14ARF stabilises an interaction between the AD and the RING domain of MDM2, which blocks E2~Ub binding and inhibits activity. There are however, some gaps - these are mostly clearly stated although some revision of the Discussion may help make this clearer.

Points to consider:

1) The authors should review the section about ARF in the introduction and see if there are ways to improve clarity.

We have clarified the ARF fragment construct used in the introduction and other sections of the manuscript, so it is not confused with full-length ARF protein. We have also included domain structure of ARF and RxFxV motif in Figure 1B for clarity.

2) The use of 're-localize' should be reviewed (note there are no page numbers!)
We have changed “re-localize” to “localized”.

3) The last section of the Introduction should be revised.
We have modified the last section in the introduction to clarify the ARF construct used in the study.

4) The MDM constructs used in this study should be clearly indicated in Figure 1A.
We have modified Figure 1A to include all MDM2 constructs used in this study.

5) Figure 1B legend and related text needs to be revised. There is discussion of co-expression in the text but lane 1 indicates GST-MDM2 is expressed on its own. The origin of lanes 2&3 is not clear. A flow diagram might help.
We have modified the legend for Figure 1B which is now Figure 1C in the revised manuscript.

6) It would be good to indicate the regions of the HSQC spectrum in Figure 1C that are blown up in 1D.
We have indicated the region with red arrow in revised Figure 1D.

7) Figures 1F &G need to be improved as the RING dimer is not obvious relative to structural features. It might be good to include a ribbon diagram, with a transparent surface (or something similar) to help orient the reader.
We have included cartoon representation of MDM2 homodimer with transparent view in the revised Figure 1G and leave Figure 1H unchanged since its orientation is the same as in Figure 1G right panel.

8) In figure 2 and the related text it is stated that the AD competes for the E2 binding site. It is clear that the AD restricts binding but not that it competes for the same binding site. Additional data should be provided to support this statement, or the text revised.
We have changed the text to “restrict binding”.

9) The text/legend related to Figure 3A should be revised in a similar way as outlined for Figure 1B. Currently it is confusing.
We have revised Figure 3A legend.

10) In Figure 3B the difference in activity seems modest, it would be good if this experiment was repeated and the data quantified, much like Figure 2C.
We have repeated the experiments in triplicates and quantified the data (see Figure 3B and 3C).

11) The results presented in Figure 3D/E suggest quite a tight interaction given the linker was cleaved. This is a nice result!
We thank the reviewer for the positive comment.

12) The structural data presented in Figure 4 might also warrant revision of the figures as indicated above.

We have revised Figure 4B and included cartoon representation of MDM2 homodimer with transparent view while leaving Figure 4C unchanged since its orientation is the same as in Figure 4B right panel.

13) It may be possible to improve/simplify the labelling of Figure 4E.

We have tried our best to simplify the labelling of Figure 4E.

14) In relation to figure 5c it is stated that when fused to p14ARF E2~ub discharge by MDM2 is nearly 'abolished'. While slower, it appears that by 5 minutes 95% of the conjugate is discharged. The authors should more accurately represent the difference in activity. It might also be useful to number the lanes and refer to them directly.

We have revised the word "abolished" to "reduced".

15) It would be helpful to enhance figure 5B so that the different constructs are more obvious - currently the domain diagram at the top is not well integrated.

We have revised Figure 5B.

16) The Discussion is quite long and in places fairly speculative. The authors should review with the goal of distinguishing models from established facts.

We have revised the discussion according to reviewer 1's comments and removed speculative discussion.

July 12, 2022

RE: Life Science Alliance Manuscript #LSA-2022-01472-TR

Prof. Danny T Huang
Cancer Research UK Beatson Institute
Garscube Estate, Switchback Road
Glasgow, Scotland G61 1BD
United Kingdom

Dear Dr. Huang,

Thank you for submitting your revised manuscript entitled "Bivalent binding of p14ARF to MDM2 RING and acidic domains inhibits E3 ligase function". We would be happy to publish your paper in Life Science Alliance pending final revisions necessary to meet our formatting guidelines.

- please address the final Reviewer #2 points
- please upload your supplementary figures as single files and add your supplementary figure legends to the main manuscript text

A. FINAL FILES:

B. MANUSCRIPT ORGANIZATION AND FORMATTING:

Sincerely,

Reviewer #1 (Comments to the Authors (Required)):

The revised manuscript by Kowalczyk and coworkers addressed questions from previous reviews by adding new experimental data and revisions to the text. The manuscript has been improved and is suitable for publication.

Reviewer #2 (Comments to the Authors (Required)):

The authors have addressed all the point raised in my initial report, as well as the points raised by the other referee. As a result the manuscript is improved and suitable for publication.

One additional point that could be improved is the colour schemes in Figures 1E and 1G. Currently cyan and magenta are used in both figures, but the cyan indicates different things and is confusing to the reader. There are numerous ways that the authors could change the colour schemes to enhance interpretation for the reader.

I also note that in figure 1H (and 4c) two helices in Ube2d2 are continuous when they are probably connected by at least one non-helical residue. This should be corrected.

Reviewer #1 (Comments to the Authors (Required)):

The revised manuscript by Kowalczyk and coworkers addressed questions from previous reviews by adding new experimental data and revisions to the text. The manuscript has been improved and is suitable for publication.

We thank the reviewer for the positive comments.

Reviewer #2 (Comments to the Authors (Required)):

The authors have addressed all the point raised in my initial report, as well as the points raised by the other referee. As a result the manuscript is improved and suitable for publication.

We thank the reviewer for the positive comments.

One additional point that could be improved is the colour schemes in Figures 1E and 1G. Currently cyan and magenta are used in both figures, but the cyan indicates different things and is confusing to the reader. There are numerous ways that the authors could change the colour schemes to enhance interpretation for the reader.

We have change the colour schemes in Figures 1E and 1G. We have also changed the colour schemes in Figures 4B and 4C to make it consistent.

I also note that in figure 1H (and 4c) two helices in Ube2d2 are continuous when they are probably connected by at least one non-helical residue. This should be corrected.

We have fixed the loop connecting the two helices in Figures 1H and 4C.

July 19, 2022

RE: Life Science Alliance Manuscript #LSA-2022-01472-TRR

Prof. Danny T Huang
Cancer Research UK Beatson Institute
Garscube Estate, Switchback Road
Glasgow, Scotland G61 1BD
United Kingdom

Dear Dr. Huang,

Thank you for submitting your Research Article entitled "Bivalent binding of p14ARF to MDM2 RING and acidic domains inhibits E3 ligase function". It is a pleasure to let you know that your manuscript is now accepted for publication in Life Science Alliance. Congratulations on this interesting work.

DISTRIBUTION OF MATERIALS:

Again, congratulations on a very nice paper. I hope you found the review process to be constructive and are pleased with how the manuscript was handled editorially. We look forward to future exciting submissions from your lab.

Sincerely,
